# A computational method for drug sensitivity prediction of cancer cell lines based on various molecular information

**Fatemeh Ahmadi Moughari[1], Changiz Eslahchi[1,2]***

**1** Department of Computer and Data Sciences, Faculty of Mathematical Sciences, Shahid Beheshti University, Tehran, Iran, **2** School of Biological Sciences, Institute for Research in Fundamental Sciences (IPM), Tehran, Iran

* ch-eslahchi@sbu.ac.ir

**Data Availability Statement:** All data and codes of the proposed method are available from the github

## Abstract

Determining sensitive drugs for a patient is one of the most critical problems in precision medicine. Using genomic profiles of the tumor and drug information can help in tailoring the most efficient treatment for a patient. In this paper, we proposed a classification machine learning approach that predicts the sensitive/resistant drugs for a cell line. It can be performed by using both drug and cell line similarities, one of the cell line or drug similarities, or even not using any similarity information. This paper investigates the influence of using previously defined as well as two newly introduced similarities on predicting anti-cancer drug sensitivity. The proposed method uses max concentration thresholds for assigning drug responses to class labels. Its performance was evaluated using stratified five-fold cross-validation on cell line-drug pairs in two datasets. Assessing the predictive powers of the proposed model and three sets of methods, including state-of-the-art classification methods, state-of-the-art regression methods, and off-the-shelf classification machine learning approaches shows that the proposed method outperforms other methods. Moreover, The efficiency of the model is evaluated in tissue-specific conditions. Besides, the novel sensitive associations predicted by this model were verified by several supportive evidence in the literature and reliable database. Therefore, the proposed model can efficiently be used in predicting anti-cancer drug sensitivity. Material and implementation are available at https://github.com/fahmadimoughari/CDSML.

## 1 Introduction

As defined by The National Research Council, Precision medicine can be used to classify the patients into subgroups that vary in response to a medical treatment [1]. Tailoring efficient treatments based on their personalized characteristics can improve the quality of therapies, avoid extra expense, and diminish undesirable side effects [2]. Therefore, predicting the sensitivity of patients toward specific treatments is an essential issue in precision medicine. Currently, the massive collection of data prepare the ground for the development of data analysis

repository (https://github.com/fahmadimoughari/CDSML).

**Funding:** The authors received no specific funding for this work.

**Competing interests:** The authors have declared that no competing interests exist.

and computational methods such as machine learning and artificial intelligence approaches [2].

Generally, the computational methods for predicting drug response have been analyzed in two ways: 1- Classification (predicting sensitive drug-cell line pairs), 2- Regression (Predicting the value of a criterion for measuring the response of a cell line toward a drug). Numerous computational approaches have been proposed to solve the classification methods that predict anti-cancer drug sensitivity using transcriptomic features of cell lines and chemical substructures of drugs.

Zhang *et al.* have designed a heterogeneous network based on drug-target associations and drug sensitivity of cell lines. It also takes drug similarities, cell line similarities, and Protein-Protein Interaction (PPI) network into account. Their method, termed HNMPRD, uses an information flow-based algorithm to predict novel sensitive pairs of cell line-drug [3]. Recently, Choi *et al.* have designed RefDNN, a computational model based on a deep neural network and myriads of ElasticNet regressors [4]. They considered a set of reference drugs and a benchmark for classifying drugs for assessing the other drugs. They predicted drug sensitivity probabilities for a specific cell line-drug pair based upon the drug's similarity to the reference set. RefDNN has the potentiality to be used for anti-cancer drug repositioning. One of the latest works in classifying cell line-drug pairs is DSPLMF [5]. This method uses a logistic matrix factorization with regularization terms. The regularization terms consider the drug similarity based on the chemical substructure and three types of cell line similarity based on gene expression profile, copy number variation, and mutation. Another similarity for cell lines was also calculated according to the response values of cell lines toward the drugs.

Furthermore, several computational regression approaches have been designed, which predicted the half-maximal inhibitory (IC50) of cell lines toward the drugs. In 2017, Wang *et al.* have suggested that the similarity of cell lines and the similarity of drugs can aid in predicting drug response values. They have proposed SRMF, a matrix factorization with regularization based on gene expression similarity of cell lines and chemical similarity of drugs [6]. They used SRMF for drug re-purposing in lung cancer cell lines. Suphavilai *et al.* have designed a recommender system, called CaDRReS, which benefits solely from cell line similarities [7]. They have shown that CaDRReS can extract meaningful information about drug mechanisms from the predicted drug responses. Wei *at al.* have introduced CDCN, which predicted drug response by inferring information from a simple network composed of cell lines and drugs [8]. CDCN yielded high-quality results despite its simple calculations. Ahmadi Moughari *et al.* have proposed ADRML, a framework for anti-cancer drug response prediction using manifold learning [9]. ARDML maps drug response values into a low-dimensional latent space and infers the drug response value for new cell line-drug pairs from the latent space. It takes several types of cell line similarities and drug similarities into account and uses them in the manifold learning procedure. They have shown that ADRML predicted good results correlated with drug pathway activities.

In recommending efficient remedies for a patient, it is essential to determine the drugs that a patient is sensitive to them; therefore, knowing drug response values itself may not give extra information in medical cases. Therefore, classifying cell line-drug pairs into sensitive/resistant pairs is a more fundamental and helpful problem than regressing their response values. On the other hand, a regression problem can be transferred to a classification problem via a thresholding technique.

In this work, as inspired by ADRML [9], we proposed CDSML, which applies the manifold learning method [10] for the classification problem using the maximum concentration threshold. Since predicting sensitive pairs is much more important than predicting drug response values, providing an efficient classification method can have a great impact in this field. We

show that using binary response as the input in the classification method leads to significantly higher performance of CDSML over ADRML.

Another contribution of this paper is introducing two novel and inclusive similarities for anti-cancer drugs besides using previously defined similarities. Due to incorporating the combination of various information in newly introduced drug similarities, using these similarities in other precision medicine models or related fields can be helpful. An interesting part of this paper is the assessment of the influence of each similarity type on the performance of CDSML. The proposed method uses a novel combination of standardization and normalization for transforming the similarity matrices into the ones with more desirable characteristics.

Moreover, this paper provides an extensive validation of using k-nearest neighbor strategy for imputing the missing values. Since the available datasets for cancer cell line sensitivity are not complete, there are numerous missing values. These assessments validate the rationality of using this technique to fill in the missing values.

In addition to the mentioned contributions, we extended the CDSML application in order to have the capability of handling missing values without imputation. Furthermore, the proposed method can be performed in various scenarios with or without similarity information and achieves highly reliable results in all scenarios. We provide a framework to assess the regression methods on classification mode.

The performance of CDSML in the classification problem is compared to the performance of off-the-shelf machine learning classifiers, as well as state-of-the-art classification and regression methods, all in the same setting. The tissue-specific results and literature evidence for the predicted sensitive pairs confirm CDSML performance.

## 2 Materials and methods

### 2.1 Data

Drug screening data and the transcriptomic data of the cell lines were obtained from the Genomics of Drug Sensitivity in Cancer (GDSC) [11] and Cancer Cell Line Encyclopedia (CCLE) [12]. The molar concentration of a drug needed for half inhibition of cell growth (IC50) as well as molecular information of cell lines in GDSC and CCLE, such as gene expression profiles, mutation, and copy number variation were downloaded using PharmacoGx R package [13]. The max concentration values for drugs were obtained from GDSC website http://cancerrxgene.org.

GDSC contains 439 drugs and 1124 cell lines; however, some IC50 values were not inserted in GDSC. Therefore, we purified the data using some pre-processing steps similar to several previous works [5, 9, 14, 15]. After applying data pre-processing steps, we obtained 555 cell lines and 98 drugs.

Moreover, CCLE dataset contains the information for 24 drugs. It should be noted that we require the max concentration values for drugs. Therefore, according to the previous studies [4], we restrict the CCLE dataset on the drugs that their max concentration values can be obtained from GDSC website. The final CCLE dataset, after applying pre-processing steps, contains 363 cell lines and 18 drugs.

In addition to the mentioned information, the PaDel descriptors of drugs in PubChem [16] were extracted using the rdkit package in Python [17]. The target proteins for drugs were gained from GDSC and DrugBank [18] databases as well as literature. The interaction information for proteins and drugs are obtained from STRING [19] and STiTCH [20] databases.

## 2.2 Pre-processing

We construct three types of matrices for assessing the gathered data. The first matrix is the response matrix which contains the drug response information. The rows of the response matrix corresponds to the cell lines and its columns are related to the drugs. The elements of response matrix are either the IC50 values or sensitivity labels. The second type of matrices is cell line feature matrices, in which the rows pertain to the cell lines and the columns represents different features. The last type of matrices is drug feature matrices, in which each row represents a drug and drug features are organized in the columns. After constructing these matrices, pre-processing steps were applied on them in order to impute the missing values, remove samples with significantly low information, calculate the cell line similarities and drug similarities, and make the data suitable for the proposed method. The pre-processing procedure includes the following four steps:

- Imputing missing values

- Converting IC50 values into binary categories

- Similarity calculation

- Standardization and normalization

These steps are elaborated in the following subsections.

**2.2.1 Imputing the missing values.** There are numerous missing values in the IC50 response matrix, cell line feature matrix based on copy number variation, and cell line feature matrix based on mutation profile. In order to remove the samples that have a significant lack of information, we omitted the drugs that have missing IC50 for more than half of the cell lines. Moreover, we removed the features of cell lines that have the missing values for the majority of cell lines. Afterward, we excluded the cell lines with missing entries for more than half of the columns in each of these matrices. Nevertheless, there were still some missing values in the matrices; therefore, we imputed these missing entries using a weighted mean of other entries. If we did not omit the samples with a significant lack of information and tried to impute all missing entries first, the obtained information would not be much reliable. The percentage of imputed data must be low enough to maintain the authenticity of data. For example, in the case of GDSC dataset, the raw IC50 matrix contains about 49% missing values. After removing the drugs and cell lines with a great extent of missing values, the obtained IC50 matrix contains only 2.7% missing pairs. Imputing such a limited fraction of data does not damage the data authenticity.

We used the following strategy to impute the remaining missing values, which is similar to the imputation procedure used in previous studies. [5, 14]. Let $E(c_i)$ be the gene expression profile of cell line $c_i$ and $I(c_i, d_j)$ be the IC50 value for cell lines $c_i$ against drug $d_j$. The missing value for $I(c, d)$ is imputed as the Eq 1.

$$I(c, d) = \sum_{i \in N(c)} \frac{I(c_i, d)D(c, c_i)}{\sum_{j \in N(c)} D(c, c_j)} \tag{1}$$

$$D(c_i, c_j) = ||E(c_i) - E(c_j)||_2^2 \tag{2}$$

where $||.||_2$ is the norm-2 and $N(c)$ is the set of indices for cell lines which are the $k$ nearest neighbors of the cell line $c$ with respect to the distance function $D$. We considered $k = 10$ for imputing missing values in GDSC and CCLE.

Moreover, the missing entries in copy number variation and mutation matrices are imputed similarly. Let $V(c, g)$ and $M(c, g)$ be the copy number variation and mutation status of gene $g$ in cell line $c$. The missing values for $V(c, g)$ and $M(c, g)$ are imputed according to Eqs 3 and 4, respectively.

$$V(c, g) = \sum_{i \in N(c)} \frac{V(c_i, g)D(c, c_i)}{\sum_{j \in N(c)} D(c, c_j)} \qquad (3)$$

$$M(c, g) = \sum_{i \in N(c)} \frac{M(c_i, g)D(c, c_i)}{\sum_{j \in N(c)} D(c, c_j)} \qquad (4)$$

It is noteworthy that gene expression profiles of cell lines are considered for calculating sample distance because the cell line feature matrix based on gene expression profiles does not contain any missing value. Therefore, the distance function $D$ can be computed for all pairs of cell lines.

**2.2.2 Convert IC50 values into binary categories.** CDSML requires discrete drug responses for classifying the cell line-drug pairs. Numerous studies have divided IC50 values into sensitive and non-sensitive classes [3–5, 14, 21]. Currently, various thresholds ($\theta$) are used to convert IC50 values into sensitive/resistance classes. In several studies, a fixed threshold is used for converting IC50 values to binary labels. For example, Brubaker *et al.* [22] used $\theta = 0.1$ and Chang *at al.* [21] used $\theta = -2$. Some studies used statistical thresholds such as drug-wise median [5, 14], mixed Gaussian distribution [3, 23], or a certain deviation from the normalized mean [24] as the threshold. While some others used reliable pharmacokinetic thresholds such as the maximum concentration of drugs ($C_{max}$) [4].

Among the various thresholds used for label assigning to drug response values, $C_{max}$ is more logical since it is based on the pharmacokinetic properties of the drug. $C_{max}$ is the maximum (peak) concentration in plasma, which is achieved by a drug. Therefore, it is evident that if a cell line requires the molar concentration of more than $C_{max}$ of a drug for half inhibition, it is resistant to the drug.

The cell lines in the GDSC database are specified as sensitive and resistant to a drug, using $C_{max}$ thresholds [11]. To clearly explain the label assignment of IC50 values, Suppose there are $m$ cell lines and $n$ drugs and $B_{m \times n}$ is a binary matrix, showing the sensitivity or resistance of cell line-drug pairs. If $B(c_i, d_j) = 1$, it denotes that cell line $c_i$ is sensitive to drug $d_j$, and resistant to it if $B(c_i, d_j) = 0$. The entries of the matrix $B$ were determined according to the following:

- If $I(c, d) < C_{max}(d)$ it is labeled sensitive, which is represented by 1.

- Otherwise, it is labeled non-sensitive, which is represented by 0

Applying $C_{max}$ threshold on the response matrix leads to labeling 59.13% of cell line-drug pairs in GDSC as sensitive pairs (i.e. 32,164 out of 54,390 pairs). Moreover, 65.1% of CCLE pairs (4,254 out of 6,534 pairs) were labeled as sensitive pairs. The remaining pairs were considered as resistant pairs.

**2.2.3 Similarity calculation.** Previous studies have frequently confirmed that similar cell lines yield similar responses to similar drugs [5–7, 25]. Therefore, the machine learning approaches can learn to predict the drug response using the similarities between cell lines and the similarities between drugs. Three types of cell line similarities and three types of drug similarities were computed for GDSC and CCLE datasets. The similarity calculation procedure is elaborated in the following.

The genomic features of cell lines are mainly characterized using gene expression profiles, copy number variation, and mutation profiles. The gene expression similarity between $c_i$ and $c_j$ cell lines is represented by $SC_E(c_i, c_j)$, which is computed by Pearson Correlation Coefficient (PCC) between the gene expression profiles of the $c_i$ and $c_j$ (See Eq 5).

$$SC_E = \frac{\sum_g (E(c_i, g) - \bar{E}(c_i))(E(c_j, g) - \bar{E}(c_j))}{\sqrt{(E(c_i, g) - \bar{E}(c_i))^2}\sqrt{(E(c_j, g) - \bar{E}(c_j))^2}} \tag{5}$$

where $E(c_i, g)$ denotes the expression of gene $g$ in cell line $c_i$ and $\bar{E}(c_i)$ represents the mean expressions of all genes in cell line $c_i$.

Moreover, $SC_V$ represent the cell line similarity based on copy number variation. Let $V(c_i, g)$ be the copy number variation of gene $g$ in cell line $c_i$ and $\bar{V}(c_i)$ mean of copy number variations of all genes for cell line $c_i$. The similarity of cell lines corresponding to copy number variation is calculated as Eq 6.

$$SC_V(c_i, c_j) = \frac{\sum_g (V(c_i, g) - \bar{V}(c_i))(V(c_j, g) - \bar{V}(c_j))}{\sqrt{(V(c_i, g) - \bar{V}(c_i))^2}\sqrt{(V(c_j, g) - \bar{V}(c_j))^2}} \tag{6}$$

In addition to the mentioned similarities between cell lines, another important feature in cancer cell lines is mutation profiles. Gene mutations play crucial functions roles in cancer development and progression [26]. Suppose $M(c_i, g)$ be a binary value, showing the mutation status of gene $g$ in cell line $c_i$, where it equals "1" if gene $g$ is mutated in cell line $c_i$ and "0" if it is wild type. $SC_M(c_i, d_j)$ represent the mutation similarity of cell lines $c_i$ and $c_j$ which is defined based on Jaccard Index (JI) of their mutation profiles.

$$SC_M(c_i, c_j) = \frac{\sum_g M(c_i, g)M(c_j, g)}{\sum_{g'}(M(c_i, g') + M(c_j, g')) - \sum_{g'}(M(c_i, g')M(c_j, g'))} \tag{7}$$

One of the frequently used similarities for drugs is the similarity of chemical substructures because chemical substructure of a drug determines its functionality up to a good extent [27–29]. The chemical substructure similarity of two drugs $d_i$ and $d_j$ is shown by $SD_S(d_i, d_j)$ which is calculated using the JI of PaDel descriptors of drugs $d_i$ and $d_j$. Let $P(d_i, l)$ be the $l$th element of the PaDel descriptor for drug $d_i$. The $SD_S(d_i, d_j)$ value is computed as Eq 8

$$SD_S(d_i, d_j) = \frac{\sum_l P(d_i, l)P(d_j, l)}{\sum_{l'}(P(d_i, l') + P(d_j, l')) - \sum_{l'}(P(d_i, l')P(d_j, l'))} \tag{8}$$

Another informative similarity of drugs can be obtained based on the interaction of drugs with other chemicals and proteins. The STiTCH database provides a comprehensive resource which presents the relationships between chemicals and proteins [20]. The relations in STiTCH are based on various sources such as experimental evidence from ChEMBL [30] PDSP Ki database [31], Protein Data Bank (PDB) [32], pathway databases including KEGG [33], Reactome [34], and NCI nature pathway interaction database [35] as well as other databases such as DrugBank [18] and MATADOR [36]. In addition to the experimental evidences from reliable databases, it uses text mining, experimentally biochemical data, gene fusion, and genomic context prediction [20]. Hence, the network obtained from STiTCH database provides an extensive insight about the drugs. The STiTCH network for CCLE drugs is illustrated in Fig 1. The oval nodes in this figure represent the drugs in CCLE and other adjacent drugs, while the circle nodes indicate the neighbor proteins. The STiTCH network for GDSC drugs is

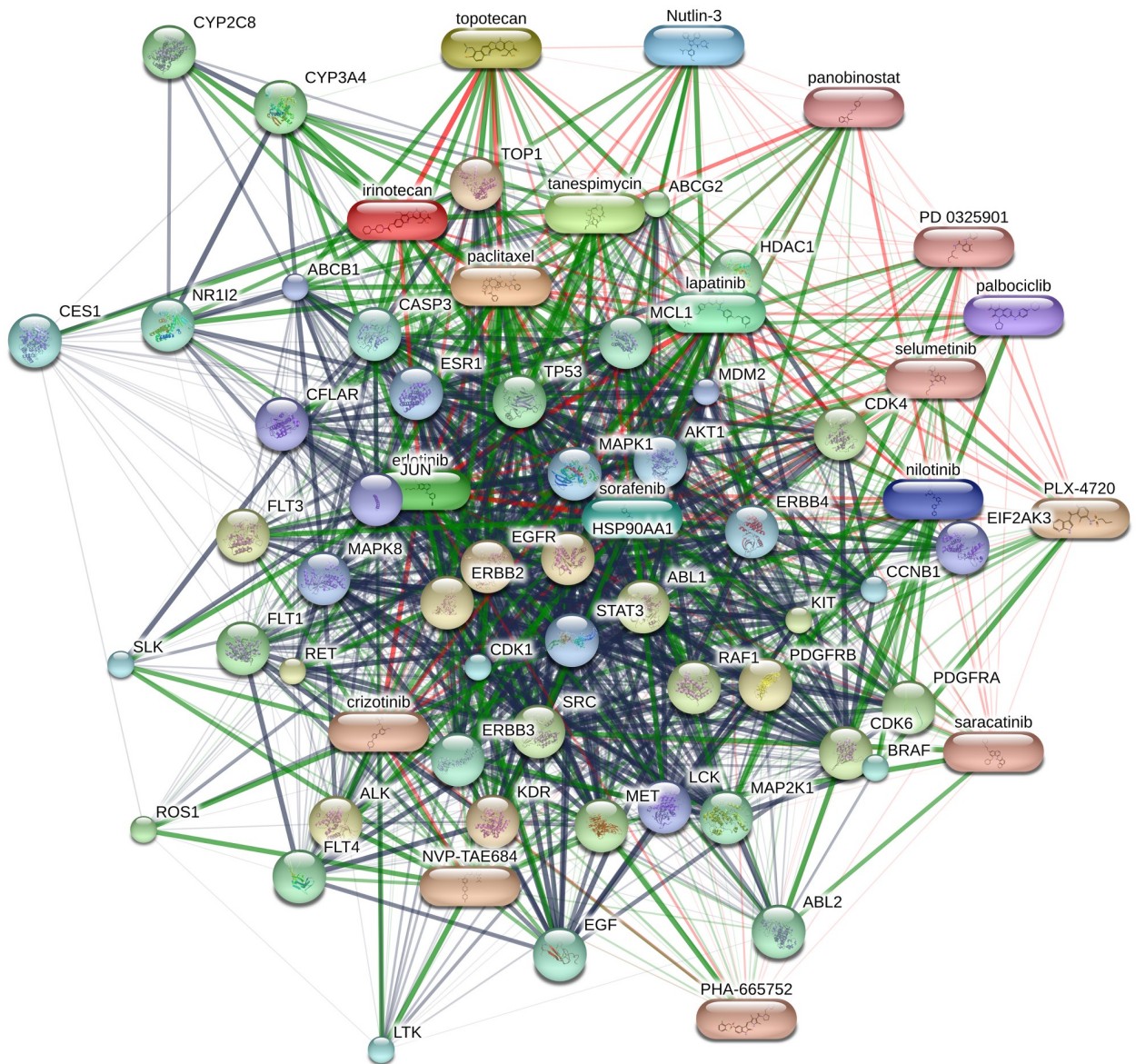

**Fig 1. STiTCH network for drugs in CCLE.**

presented in the S2 File (S1 Fig in S2 File). The weights of edges between nodes in STiTCH network for GDSC and CCLE drugs are presented in S1 and S2 Tables in S1 File, respectively.

We computed drug similarities based on STiTCH network ($SD_N$) according to the Eq 9, where $N_i$ represents the immediate neighbors (both protein neighbors and chemical neighbors) of drug $d_i$. Therefore the JI of the neighbor nodes in STiTCH network is used as the second similarity of drugs.

$$SD_N(d_i, d_j) = \frac{|N_i \cap N_j|}{|N_i \cup N_j|} \tag{9}$$

In addition to the above drug similarities, we calculated another similarity based on PPI network of target proteins. To this aim, we obtained drug targets and considered the target

proteins for all drugs in the dataset as the *target protein set*. The *target protein set* for GDSC and CCLE are presented in S3 and S4 Tables in S1 File, respectively. Afterwards, we acquired PPI network for the *target protein set* from STRING database [19]. Fig 2 represents the STRING PPI network for *target protein set* of CCLE drugs. The STRING PPI network for GDSC drugs is presented in the S2 File (S2 Fig in S2 File). It is noteworthy that STRING database uses various types of data such as Gene ontology terms, pathways experimental evidence, and text mining features to compute the interactions between proteins. Hence, the weights of edges in STRING PPI network are the combination of various evidence. The weights of edges between nodes in STiTCH network for GDSC and CCLE drugs are presented in S5 and S6 Tables in S1 File, respectively.

To compute the PPI-based similarity of drugs ($SD_P$) we constructed a bipartite graph for each pair of drugs. To explain clearly, suppose that $TP_i$ and $TP_j$ are the sets of target proteins for drugs $d_i$ and $d_j$, respectively. Bipartite PPI graph $G(i, j)^{PPI}$ is constructed such that the set of nodes in one part is $TP_i$, the set of nodes in another part is $TP_j$ and the edges are defined based on STRING PPI network. For example, assume that that $TP_i = \{P_{i1}, P_{i2}, P_{i3}\}$ is the target proteins of drug $d_i$, while $TP_j = \{P_{j1}, P_{j2}, P_{j3}, P_{j4}\}$ is the target proteins of drug $d_j$. The bipartite graph $G(i, j)^{PPI}$ is illustrated in Fig 3. The weights PPI edges between two parts in this graph are set by the weights of PPI in STRING network. Afterwards, we applied the maximum weighted matching algorithm [37] on this bipartite graph and consider the summation of weights of matching edges as the PPI-based similarity of drug pairs. This similarity shows the extent of accordance between the set of target proteins of two drugs. Therefore, $SD(d_i, d_j)$ is a high value, if the set of target proteins for $d_i$, $d_j$ match highly.

**2.2.4 Standardization and normalization of similarity matrices.** After computing all similarities, we standardized the similarity matrices in order to ensure that all similarity values range from 0 to 1. Since all similarities were computed based on PCC or JI, the similarity values range from -1 to 1; therefore, this standardization transforms the values in the range of [0, 1]. To clearly explain the standardization process, let $S(i, j)$ be an entry in a similarity matrix. Its standardized value is represented by $\hat{S}(i, j)$ and is computed according to Eq 10.

$$\hat{S}(i, j) = \frac{S(i, j) + 1}{2} \qquad (10)$$

It is notable that performing standardization on similarity matrices does not change the sorting of distances between samples because this is a linear transform.

In the next step, we normalized the standardized similarity matrices using the symmetric normalized Laplacian [38]. The symmetric normalized Laplacian is a well-defined transform of the similarity matrix with several favorable algebraic and spectral characteristics such as being positive definite and diagonally dominant [39]. Moreover, its prevalence use in other problems such as spectral clustering [40] and drug target interaction prediction [41] justifies that using symmetric normalized Laplacian matrix represents the similarity of samples and shows the structural properties in a better way [42]. For each similarity matrix $S$, the normalized similarity matrix $\mathcal{S}$ is obtained as Eq 11.

$$\mathcal{S} = D^{-1/2}(D - S)D^{-1/2} \qquad (11)$$

In this equation, $D$ is a diagonal matrix and $D_{i,i} = \sum_j S_{i,j}$. All diagonal elements in $D$ are non-zero. Hence, $D^{-1/2}$ is a diagonal matrix and its entries are reverse values of the square root of elements in $D$. It should be noted that this normalization does not affect the correlation between various types of similarity matrices. On the other hand, applying this normalization improves the model speed and leads to quicker convergence of the CDSML.

**Fig 2. STRING network for drugs in CCLE.**

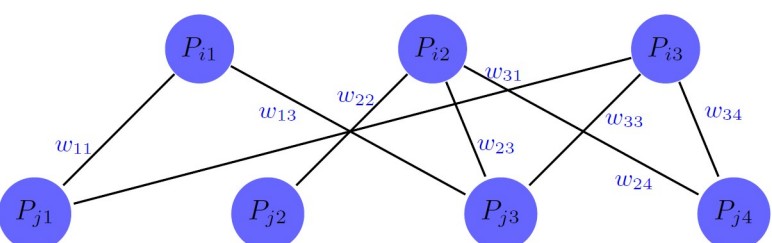

**Fig 3. Schematic representation of the bipartite graph between the set of target proteins for a drug pair.**

## 2.3 Applying classification method

The classification process is inspired from ADRML method [9], but with binary response matrix $B$, rather than IC50 value matrix $I$. Moreover, it uses a threshold to convert obtained results into binary labels. This method has several steps, including:

1. Decompose $B_{m \times n}$ matrix into $X_{m \times k}$ and $Y_{n \times k}$, such that $B \approx XY^T$

    (a).  Initialize $X^{(0)}$ and $Y^{(0)}$ randomly.

    (b).  Compute a loss function defined as the summation of mean square error (MSE), similarity conservation terms and regularization terms.

    (c).  Update $X$ and $Y$ matrices according to the Newton's method.

    (d).  Repeat steps 1.b and 1.c until $X$ and $Y$ matrices converge.

2. Decompose $B^T$ into two latent matrices $W$ and $Z$ similar to step 1.

3. Compute $\tilde{B} = \dfrac{1}{2}(XY^T + WZ^T)$ as the predicted sensitivity matrix.

4. Use a threshold to convert $\tilde{B}$ into a binary matrix $\hat{B}$.

    These steps are shown in Fig 4 and explained elaborately in the following.

    **2.3.1 Compute loss function.**  To classify the cell line-drug pairs, $B_{m \times n}$ is decomposed into two latent matrices $X_{m \times k}$ (cell line latent matrix) and $Y_{n \times k}$ (drug latent matrix) with lower rank. The decomposition must satisfy the following constraints:

- Decomposition Mean Square Error (MSE): The multiplication of latent matrices $XY^T$ must be an appropriate estimation of binary response matrix $B$.

- Regularization: The elements of $X$ and $Y$ matrices should not grow excessively.

- Cell line similarity conservation: The similar cell lines must have not too far latent row vectors in $X$ matrix.

- Drug similarity conservation: The similar drugs must have not too far latent row vectors in $Y$ matrix.

    Considering all the above constraints, the loss function is defined according to Eq 12.

$$
\begin{aligned}
Loss \quad &= \frac{1}{2}\sum_{i,j}(B(i,j) - X(i)Y(j)^T)^2 + \frac{\alpha}{2}\left(\sum_i ||X(i)||^2 + \sum_j ||Y(j)||^2\right) \\
&+ \frac{\beta}{2}\left(\sum_{i,j}||X(i) - X(j)||^2 SC(i,j) + \sum_{i,j}||Y(i) - Y(j)||^2 SD(i,j)\right) (12)
\end{aligned}
$$

where $\alpha$ and $\beta$ are the regularization and similarity conservation coefficients. $X(i)$ and $Y(i)$ denotes the $i$th rows in $X$ and $Y$ matrices, respectively. The symbol $SC$ in Eq 12 can be substituted by $SC_E$, $SC_C$, or $SC_M$. Two latent matrices $X$ and $Y$ were updated using Newton's method to minimize the loss function iteratively. $X^{(0)}$ and $Y^{(0)}$ were initialized randomly and

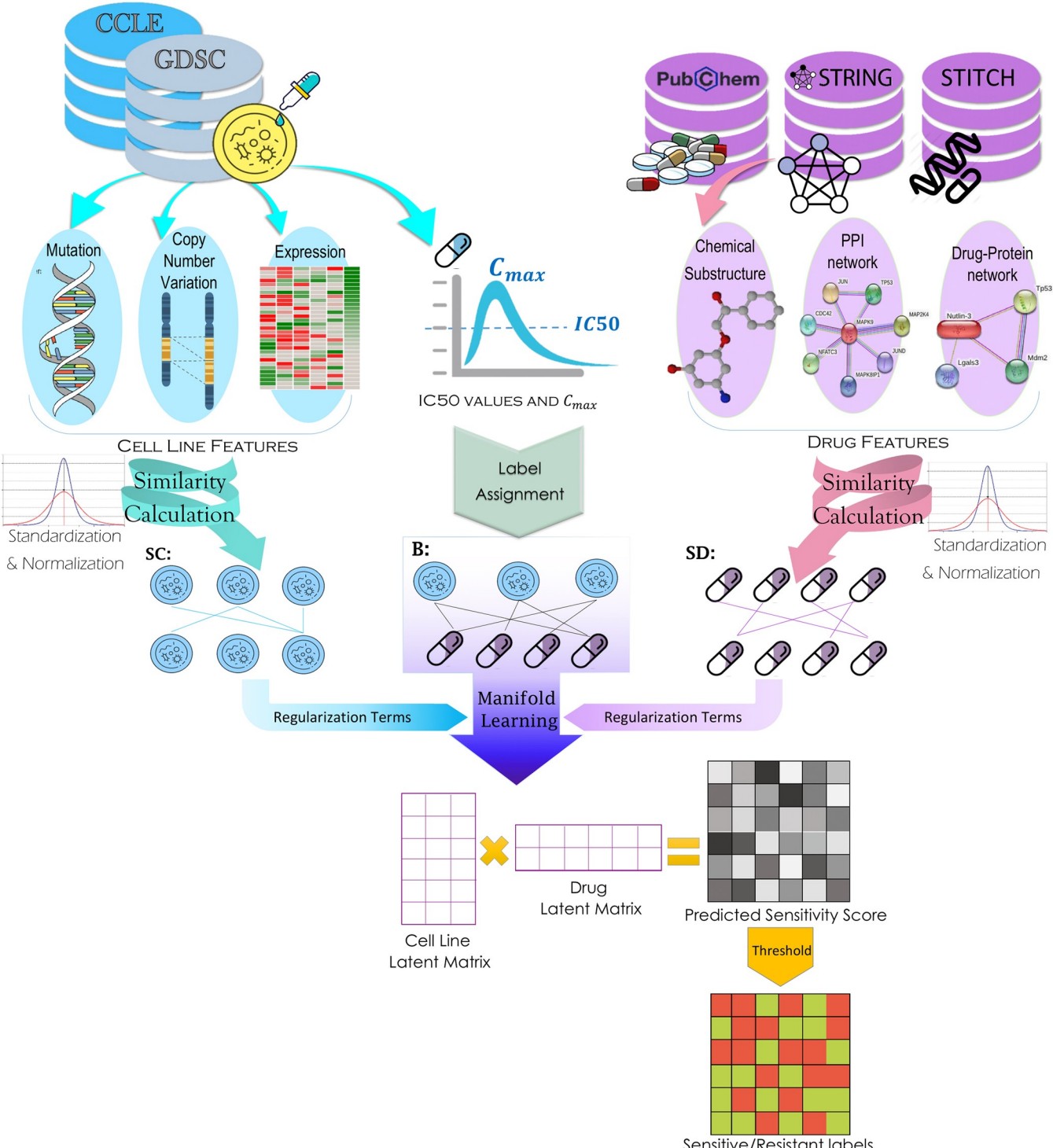

**Fig 4. The overall framework of the proposed method.** The features cell lines, IC50 and $C_{max}$ values were obtained from GDSC. Besides, drug substructures were downloaded from PubChem, STiTH network from STiTCH datbase, and PPI network from STRING database. The sensitivity associations between cell line-drug pairs were specified and used as the objective of manifold learning. The cell line similarities and drug similarities were calculated, standardized, and normalized. These similarities were considered as the regularization terms in manifold learning. The output of manifold learning is a predicted score matrix that assigns cell line-drug pairs into sensitive resistant classes by a threshold.

afterwards, $X^{(k)}$ and $Y^{(k)}$ were updated using the rules defined in Eqs 13 and 14, respectively.

$$X^{(k+1)} = X^{(k)} - \frac{\nabla_{X^{(k)}} Loss}{\nabla^2_{X^{(k)}} Loss} \qquad (13)$$

$$Y^{(k+1)} = Y^{(k)} - \frac{\nabla_{Y^{(k)}} Loss}{\nabla^2_{Y^{(k)}} Loss} \qquad (14)$$

These matrices were updated until they do not change significantly in two subsequent iterations. Specifically, when $||X^{(k+1)} - X^{(k)}|| + ||Y^{(k+1)} - Y^{(k)}|| < 0.01$, the convergence criteria is met. The detailed formulae for updating latent matrices are described in S2 File.

**2.3.2 Predicting sensitivity/resistant labels.** As described above, we decomposed $B$ into two latent matrices and after convergence, the last estimated latent matrices $X^{(k)}$ and $Y^{(k)}$ were multiplied to estimate $B$:

$$\tilde{B}_1 = X^k Y^{k^T} \qquad (15)$$

Then, $B^T$ is decomposed into $W_{m \times k}$ and $Z_{n \times k}$ using the described method. This is done due to the fact that the predicted labels for samples $(c_i, d_j)$ and $(d_j, c_i)$ must be equal. After the convergence, $B^T$ is estimated using the following equation:

$$\tilde{B}_2 = W^k Z^{k^T} \qquad (16)$$

The predicted sensitivity matrix ($\tilde{B}$) was calculated as the average of $\tilde{B}_1$ and $\tilde{B}_2$. It should be noted that the estimated matrix $\tilde{B}$ is not binary valued; thus, we converted it to a binary-valued matrix $\hat{B}$ using a threshold. So that the cell line-drug pairs were assigned to sensitive/resistant classes. The computation of best threshold is explained in Section 2.4.

## 2.4 Evaluation criteria

The predictive performance of models was assessed using stratified five-fold cross-validation on cell line-drug pairs. To this aim, the set of all cell line-drug pairs were partitioned randomly to five subsets of equal sizes such that the fraction of sensitive over resistant pairs was almost equal in all subsets. Four subsets were considered as the training data and the evaluation criteria were computed on the remaining subset. This procedure was iterated for each subset and the criteria were averaged over the iterations. The stratified five-fold cross-validation was repeated 30 times in order to prevent bias in partitioning the dataset. The most prevalent evaluation criteria for classification problems are defined in the following:

$$Recall = \frac{TP}{TP + FN} \qquad (17)$$

$$Precision = \frac{TP}{TP + FP} \qquad (18)$$

$$F1 - score = \frac{2 \times Recall \times Precision}{Recall + Precision} \qquad (19)$$

$$Accuracy = \frac{TP + TN}{TP + TN + FP + FN} \qquad (20)$$

Where *TP*, *TN*, *FP* and *FN* stands for true positive, true negative, false positive, and false negative, respectively. These statistics are defined in Table 1.

It should be noted that the mentioned criteria are threshold dependent. *AUPR* and *AUC* are more reliable criteria that are independent of threshold values. *AUPR* is the area under the plot of *Precision* versus *Recall* at various thresholds. *AUC* is the area under the ROC Curve which plotted *Recall* against $FPR = \dfrac{FP}{FP + TN}$ at different threshold values.

The best threshold converting $\tilde{B}$ estimated scores to binary labels were determined using *Precision* − *Recall* curve. The threshold value related to the elbow-point in this curve is considered as the best threshold because at this threshold, there is a good balance between *Precision* and *Recall*.

## 2.5 Variations of CDSML

The application of CDSML can be extended in order to be performed in various scenarios. The following subsections includes two variations of CDSML.

**2.5.1 Performing on the response matrix with missing values.** CDSML can handle the binary response matrix *B* without imputing missing values. To this aim, it is required to alter the Eq 12 such that it computes the loss function only for known pairs. Eq 21 is capable of handling the response matrix which contains missing values.

$$
\begin{aligned}
Loss^{(Missing)} \quad &= \frac{1}{2} \sum_{i,j \notin Missing} (B(i,j) - X(i)Y(j)^T)^2 + \frac{\alpha}{2}\left( \sum_i ||X(i)||^2 + \sum_j ||Y(j)||^2 \right) \\
&+ \frac{\beta}{2}\left( \sum_{i,j} ||X(i) - X(j)||^2 SC(i,j) + \sum_{i,j} ||Y(i) - Y(j)||^2 SD(i,j) \right) (21)
\end{aligned}
$$

where *Missing* denotes the set of missing pairs. If we use Eq 21 instead of Eq 12, the method is able to be performed without applying the imputation step for missing IC50 values. The detailed formula for updating latent matrices in this version is explained in the S2 File. It should be noted that using these formula works also for response matrix without missing. For example, when the missing values are imputed, the set of Missing is empty; therefore, in that case the loss terms are calculated for all pairs.

**2.5.2 Using double, single, or no similarity matrices.** One can perform CDSML in three different scenarios based on similarity usage:

- Double similarity: using both *SC* and *SD* similarity matrices

- Single similarity: using either *SC* or *SD* similarity matrices

- No similarity: using no similarity matrix.

**Table 1. The confusion table for defining classification statistics.**

| | | Real labels | |
| --- | --- | --- | --- |
| | | **Sensitive** | **Resistant** |
| Predicted labels | Sensitive | TP | FP |
| | Resistant | FN | TN |

To this aim, the loss function in Eq 21 must be modified. If only *SC* is ignored, the loss function will be changed to the Eq 22.

$$
\begin{aligned}
Loss^{(SD)} \quad &= \frac{1}{2} \sum_{i,j \notin Missing} (B(i,j) - X(i)Y(j)^T)^2 + \frac{\alpha}{2}\left( \sum_i ||X(i)||^2 + \sum_j ||Y(j)||^2 \right) \\
&+ \frac{\beta}{2}\left( \sum_{i,j} ||Y(i) - Y(j)||^2 SD(i,j) \right) (22)
\end{aligned}
$$

If only *SD* is ignored, the loss function will be changed to the Eq 23.

$$
\begin{aligned}
Loss^{(SC)} \quad &= \frac{1}{2} \sum_{i,j \notin Missing} (B(i,j) - X(i)Y(j)^T)^2 + \frac{\alpha}{2}\left( \sum_i ||X(i)||^2 + \sum_j ||Y(j)||^2 \right) \\
&+ \frac{\beta}{2}\left( \sum_{i,j} ||X(i) - X(j)||^2 SC(i,j) \right) (23)
\end{aligned}
$$

If both *SC, SD* are ignored, the loss function will be changed to the Eq 24.

$$
Loss^{(No\ sim)} = \frac{1}{2} \sum_{i,j \notin Missing} (B(i,j) - X(i)Y(j)^T)^2 + \frac{\alpha}{2}\left( \sum_i ||X(i)||^2 + \sum_j ||Y(j)||^2 \right) \quad (24)
$$

It is notable that, in each scenario, the equations for updating latent matrices will be adjusted based on the related loss functions. The related equations for updating latent matrices in each scenario is presented in the S2 File.

It is notewothy that both variations explained in Sections 2.5.1 and 2.5.2 can be performed simultaneuosly. In other words, when the user wants to perform CDSML in each of similarity scenarios, the response matrix may contain missing values or not. Because the formula used in various similarity scenarios can ignore missing values if there are any.

## 3 Results

In this section, we present the results of evaluating CDSML and compring its performance with other methods.

### 3.1 Tuning hyper-parameters

We tuned CDSML hyper-parameters using grid search on different values of hyper-parameters on GDSC dataset using $SC_E$, $SD_S$. Then we used the same hyper-parameters for CCLE or when using other similarities. We considered $\alpha, \beta \in \{0.125, 0.25, 0.5, 1, 1.5, 2, 2.5, \cdots, 8\}$ and $K = k' * min(m, n)$, where $k' \in \{0.1, 0.2, \cdots, 0.9\}$. The best values for hyper-parameters was determined based on *AUC* criterion because this criterion is independent from threshold value and assess the model more extensively. The best hyper-parameters for CDSML was $\alpha = 3.5$, $\beta = 4.5$, and $k' = 0.7$.

### 3.2 The performance of CDSML

CDSML predicts drug sensitivity according to the cell line similarity and drug similarities. We calculated three types of cell line similarities based on gene expression, mutation profile, and copy number variation. Each of these cell line similarities can be utilized as *SC* in CDSML similarity conservation terms. Furthermore, we computed three types of drug similarities based on chemical substructure, STiTCH network and target PPI, each of which can be considered

as *SD* in CDSML similarity conservation terms. We analyzed the impact of using different types of similarities on the classification performance of CDSML. Table 2 represents the CDSML performance on GDSC dataset in three scenario, namely, double similarity, single similarity, and no similarity.

We discuss about each of three scenarios in the following.

- Discussing about "No Similarity" scenario:
  The results in the first row of Table 2 indicate that CDSML gained high performance even using no similarities. CDSML performance using no similarity confirmed that the matrix factorization used in CDSML is efficient itself without using other extra information. The CDSML results using no similarity outperforms most of state-of-the-art methods mentioned in the paper.

- Discussing about "Single Similarity" scenario:
  Based on this table, one can conclude that using a single similarity matrix makes about 6% improvements of all criteria compared to the case of no similarities. Moreover, the performance of CDSML on various types of SC or various types of SD leads to almost equal performance, i.e. the calculated criteria for all executions on a single similarity matrix are the same. For example, using only the copy number variation similarity of cell lines leads to *AUC* of 0.9071 and *AUPR* of 0.9353, while using only the PPI similarity of drugs also leads to the same values of *AUC* and *AUPR*. Therefore, the impact of all similarity matrix on the CDSML performance is almost equal. Thus, CDSML yielded highly accurate and robust results.

- Discussing about "Double Similarity" scenario:
  Moreover, by comparing the results in single and double similarity scenarios, it can be inferred that the performance of CDSML using both SC and SD make subtle improvement in comparison to the single similarity scenario.

The performance of CDSML on CCLE dataset using different scenarios including double similarity, single similarity, and no similarity are provided in S2 File. Further evaluations on CDSML were conducted using cell line similarity based on gene expression and chemical substructure similarity of drugs.

## 3.3 Evaluation of predicted and imputed values for missing pairs

In order to show the rationality of imputed values in response matrix using Eq 1, one can compare the predicted labels by CDSML for missing values (denoted by $Pred_L^{(CDSML)}$) with imputed labels using Eq 1 (denoted by ($Impute_L^{(Eq.1)}$). x *L* shows that the vector consists of binary labels. $Pred_L^{(CDSML)}$ is computed by executing CDSML on the binary response matrix with missing values (not applying imputation procedure), while considering all known pairs as training samples and all missing pairs as test samples. On other hand, $Pred_L^{(CDSML)}$ is computed by imputing missing values using Eq 1 and then converting the imputed values $Impute_C^{(Eq.1)}$ (index *C* denotes that the vectors contain the continuous values) to the imputed labels $Impute_L^{(Eq.1)}$ by comparing imputed IC50 values with max concentration thresholds of drugs. Since both vectors are binary, computing *Accuracy*, *F*1 − *score*, *Precision*, and *Recall* in addition to JI, cosine similarity, and cross entropy are meaningful and can give us an extensive comparison of these two vectors. *Accuracy*, *F*1 − *score*, *Precision*, and *Recall* are computed as defined in Section 2.4. The JI, cosine similarity, and binary cross entropy of two vectors *X*, *Y* are defined in

**Table 2. The performance of CDSML on GDSC dataset using different scenarios including double similarity, single similarity, and no similarity.** The best value of each criterion is shown in bold.

| | SC | SD | AUC | AUPR | Accuarcy | F1-score | Precision | Recall |
|---|---|---|---|---|---|---|---|---|
| No sim | - | - | 0.8542 | 0.8846 | 0.7812 | 0.8291 | 0.7706 | 0.8974 |
| Single similarity | $SC_E$ | - | 0.9147 | 0.9372 | 0.8359 | 0.8653 | 0.841 | 0.891 |
| | $SC_M$ | - | 0.9148 | 0.9373 | 0.8356 | 0.8652 | 0.8401 | 0.892 |
| | $SC_V$ | - | 0.9071 | 0.9353 | 0.8288 | 0.8636 | 0.8346 | 0.8948 |
| | - | $SD_S$ | 0.9071 | 0.9353 | 0.8288 | 0.8636 | 0.8346 | 0.8951 |
| | - | $SD_N$ | 0.9072 | 0.9354 | 0.83 | 0.8637 | 0.8392 | 0.8899 |
| | - | $SD_P$ | 0.9071 | 0.9353 | 0.8292 | 0.8638 | 0.8357 | 0.8943 |
| Double similarity | $SC_E$ | $SD_S$ | **0.9158** | 0.9373 | **0.8388** | 0.8714 | **0.8426** | 0.9026 |
| | $SC_E$ | $SD_P$ | 0.9157 | 0.9373 | 0.8354 | 0.8714 | 0.8423 | 0.903 |
| | $SC_E$ | $SD_N$ | 0.9157 | **0.9398** | **0.8388** | **0.8715** | 0.8422 | **0.9031** |
| | $SC_M$ | $SD_S$ | 0.9148 | 0.9373 | 0.8357 | 0.8652 | 0.8402 | 0.8918 |
| | $SC_M$ | $SD_P$ | 0.9147 | 0.9373 | 0.8354 | 0.8652 | 0.8392 | 0.8929 |
| | $SC_M$ | $SD_N$ | 0.9147 | 0.9373 | 0.8350 | 0.8652 | 0.8388 | 0.8949 |
| | $SC_V$ | $SD_S$ | 0.9157 | 0.9397 | **0.8388** | 0.8714 | **0.8426** | 0.9026 |
| | $SC_V$ | $SD_N$ | 0.9157 | **0.9398** | 0.8387 | 0.8714 | 0.8424 | 0.9028 |
| | $SC_V$ | $SD_P$ | 0.9147 | 0.9372 | 0.8355 | 0.8653 | 0.8390 | 0.8934 |

Eqs 25–27.

$$JI(X, Y) = \frac{\sum_i X(i)Y(i)}{\sum_i (X(i) + Y(i)) - \sum_i X(i)Y(i)} \quad (25)$$

$$Cosine\ Similarity(X, Y) = \frac{\sum_i X(i)Y(i)}{||X||_2 ||Y||_2} \quad (26)$$

$$Cross\ Entropy(X, Y) = \frac{-1}{n} \sum_i [X(i)\ \log\ Y(i) + (1 - X(i))\ \log\ (1 - Y(i))] \quad (27)$$

Note that we need to compute true positive, true negative, false positive, and false negative sample to compute the *recision* and *Recall*. Therefore, we must consider one of $Pred_L^{(CDSML)}$ or $Impute_L^{(Eq.1)}$ as the ground truth labels. Note that *Precision* value computed by considering $Pred_L^{(CDSML)}$ as the ground truth equals to the *Recall* value computed by considering $Impute_L^{(Eq.1)}$ as the ground truth, and vice versa. All mentioned metrics lies in the range of [0, 1]. The higher values of these metrics (except cross entropy) are more satisfactory, while the lower values of cross entropy are more favorable.

The computed metrics by considering $Pred_L^{(CDSML)}$ as the ground truth are shown in Table 3. According to the significantly low value of cross-entropy as well as the high value of other metrics, one can conclude that $Impute_L^{(Eq.1)}$ and $Pred_L^{(CDSML)}$ are considerably similar. Consequently, the imputed values for missing pairs are reasonable.

**Table 3. Comparison of $Impute_L^{(Eq.1)}$ and $Pred_L^{(CDSML)}$ on GDSC dataset.**

| Accuracy | F1-score | Precision | Recall | JI | Cosine similarity | Cross entropy |
|---|---|---|---|---|---|---|
| 0.718 | 0.764 | 0.9297 | 0.6484 | 0.6181 | 0.7764 | 0.0065 |

**Table 4. Comparison of $Impute_C^{(Eq.1)}$ and $Pred_C^{(Zhang)}$ on GDSC dataset.**

| RMSE | $NRMSE_{(Impute)}$ | $NRMSE_{(Zhang)}$ | MAE |
|---|---|---|---|
| 0.04369 | 0.0065 | 0.0081 | 1.45 |

To further justify the rationality of imputed values, we compared the imputed IC50 values using Eq 1 ($Impute_C^{(Eq.1)}$) with the predicted IC50 values by an state-of-the-art method. The method proposed by Zhang *et al.* predicts IC50 value uses a dual layer network which is similar to the idea used in Eq 1 [25], while having some differences. Moreover, Zhang *et al.* have shown that the method predicts reliable IC50 values for missing pairs by providing biological evidence for the missing IC50 values of three MEK inhibitor drugs. Thus, it is interesting to compare the IC50 values imputed using Eq 1 ($Impute_C^{(Eq.1)}$) with the predicted IC50 values by Zhang *et al.* method($Pred_C^{(Zhang)}$). To compare these two continuous vectors, regression criteria such as Root Mean Square Error (RMSE), Normalized Root Mean Square Error (NRMSE), and Mean Absolute Error (MAE) can be computed. RMSE, NRMSE, and MAE for two vectors *X, Y* are defined in Eqs 28–30.

$$RMSE(X, Y) = \sqrt{\frac{\sum_i (X(i) - Y(i))^2}{n}} \tag{28}$$

$$NRMSE_X(X, Y) = \frac{RMSE}{\max_i X(i) - \min_i X(i)} \quad NRMSE_Y(X, Y) = \frac{RMSE}{\max_i Y(i) - \min_i Y(i)} \tag{29}$$

$$MAE(X, Y) = \frac{\sum_i |X(i) - Y(i)|}{n} \tag{30}$$

It should be noted that RMSE, NRMSE, and MAE ranges from 0 to infinity. So, lower values of them shows that the *X, Y* are closer to each other. The computed metrics for comparing these two vectors are presented in Table 4 The computed metrics show that the imputed IC50 values in this paper are very close to the Zhang *et al.* predicted IC50 values. Additionally, the comparison of imputed values with Zhang *et al.* predictions in binary mode as well the comparison of CDSML predictions with Zhang *et al.* predictions on missing pairs are provided in the S2 File.

An interesting idea is to use Zhang *et al.* predicted IC50 values ($Pred_C^{(Zhang)}$) for filling missing values in response matrix. To this aim, we converted continuous values of ($Pred_C^{(Zhang)}$) into ($Pred_L^{(Zhang)}$) with binary entries by comparing the max concentration with $Pred_C^{(Zhang)}$. In other words, we used Zhang *et al.* predicted labels ($Pred_L^{(Zhang)}$) instead of labels computed by Eq 1; i.e. $Impute_L^{(Eq.1)}$ for filling the missing values in the response matrix. We then performed CDSML manifold learning on the obtained response matrix. Let us denote this version as CDML-Zhang and compared its results with CDSML using a stratified 5-fold cross-validation. Table 5 shows the comparison between CDSML and CDSML-Zhang. It can be seen that the

**Table 5. Comparison of CDSML and CDSML-Zhang performance on GDSC dataset.** The assessments were done by averaging 30 repetitions of stratified five-fold cross-validation. The highest value of each criterion is shown in bold.

| Method | AUC | AUPR | Accuracy | F1-score | Precision | Recall |
|---|---|---|---|---|---|---|
| CDSML | **0.9157** | **0.9398** | **0.8388** | **0.8715** | **0.8422** | **0.9031** |
| CDSML-Zhang | 0.912 | 0.937 | 0.836 | 0.870 | 0.84 | 0.896 |

evaluation criteria computed for both versions are so close to each other. However, CDSML leads to better results. These comparisons additionally validates using Eq 1 for imputing missing entries of response matrix.

To sum up all validation scenarios in this section and the related sections in S2 File, one can conclude that the results of all validations confirm that the imputed values using Eq 1 are reasonable and leads to the improvement in results.

## 3.4 Comparisons with classification methods

To compare the predictive performance of CDSML with other state-of-the-art classification methods, we used available implementations for HNMPRD [3], RefDNN [4], and DSPLMF [5]. In order to have a fair comparison, we evaluated these methods using 30 repetitions of stratified five-fold cross-validation on cell line-drug pairsfor GDSC and CCLE. These methods cover a variety of classification methodology and labeling thresholds. The methodology of HNMPRD, RefDNN, and DSPLMF are based on information flow, deep neural network, and matrix factorization, respectively. Moreover, the thresholds used in HNMPRD, RefDNN, and DSPLMF to convert IC50 values to sensitive/resistance labels are mixed Gaussian distribution, $C_{max}$, and drug-wise median, respectively.

A comparison between CDSML and other state-of-the-art methods for category classification is represented in Table 6. Considering GDSC dataset, HNMPRD obtained high *Recall*, but low values in other criteria. RefDNN showed satisfying performance according to all and DSPLMF obtained reasonable results according to *Recall*, but not high quality results with respect to other criteria. CDSML outperforms other classification methods by achieving much higher values of all criteria than other methods. It improves the results of RefDNN by almost 2% in *AUC*, 7% in *AUPR*, 2% in *Accuracy*, and 9% in *F1 − score*.

Considering CCLE dataset, HNMPRD again obtained high *Recall*, but low values in other criteria. RefDNN and DSPLMF revealed acceptable performance. Consequently, the CDSML performance is significantly higher than other state-of-the-art classification methods and improves the best values of *AUC*, *AUPR*, *Accuracy* and *F1 − score* by more than 10%.

To fully compare the CDSML performance with classification methods, we compared its results with six off-the-shelf classification methods covering diverse methodologies: Gaussian Naiive Bayes (GNB), logistic regression (LR), random forest (RF), multi-layer perception (MLP), adaptive boosting (ADA), and K-nearest neighbor (KNN). The implementations of all these methods were conducted using the Scikit-learn python package [43]. It should be noted that the feature vector for each pair of cell line $c_i$ and drug $d_j$ was constructed by concatenating the $i$th row of *SC* and $j$th column of *SD*. Moreover, the $C_{max}$ was used to label the cell line-drug pairs. The best hyper-parameter values were specified using grid search. The set of evaluated hyper-parameters and the best values of hyper-parameters for all methods are presented in Table 7. The best values of hyper-parameters were tuned using grid search and considering *AUC* criterion.

Table 8 provides the comparison between CDSML and off-the-shelf classification methods. On both GDSC and CCLE, all methods showed good performance in classification of anti-cancer drug sensitivity; however, the least and highest values of criteria belongs to GNB and RF, respectively. On top of them, CDSML achieved the most accurate results and outperforms other methods according to all criteria, except *Precision*. Nevertheless, its *Precision* is not too far from the best *Precision*.

It is noteworthy that the computed criteria for machine learning models are significantly higher than the computed criteria for state-of-the-art classification and regression methods. The tuned hyper-parameters for machine learning models turn them into efficient and potent

**Table 6. Comparison of CDSML's performance with state-of-the-art classification methods' performance on GDSC and CCLE.** The assessments were done by averaging 30 repetitions of stratified five-fold cross-validation. The highest value of each criterion is shown in bold.

| Dataset | Method | AUC | AUPR | Accuracy | F1-score | Precision | Recall |
|---------|--------|------|------|----------|----------|-----------|--------|
| GDSC | CDSML | **0.9157** | **0.9398** | **0.8388** | **0.8715** | **0.8422** | 0.9031 |
| GDSC | HNMPRD | 0.5728 | 0.6121 | 0.6088 | 0.7494 | 0.6032 | **0.9894** |
| GDSC | RefDNN | 0.9013 | 0.8753 | 0.8219 | 0.7828 | 0.8114 | 0.7583 |
| GDSC | DSPLMF | 0.7350 | 0.7218 | 0.6407 | 0.7096 | 0.5947 | 0.8797 |
| CCLE | CDSML | **0.9514** | **0.977** | **0.8989** | **0.9201** | **0.9485** | 0.8934 |
| CCLE | HNMPRD | 0.5817 | 0.7228 | 0.6527 | 0.7889 | 0.6528 | **0.9967** |
| CCLE | RefDNN | 0.6951 | 0.8200 | 0.6796 | 0.7825 | 0.7109 | 0.8713 |
| CCLE | DSPLMF | 0.8148 | 0.6944 | 0.7405 | 0.6667 | 0.5906 | 0.7668 |

models that outperform other state-of-the-art methods. These findings are in agreement with the similar findings in RefDNN paper [4].

Figs 5 and 6 demonstrates the ROC curve and *Precision − Recall* curve for all of the classification methods on GDSC dataset, respectively. As it is shown the *AUC* and *AUPR* values for CDSML are 0.9221 and 0.943, respectively which are superior to the *AUC* and *AUPR* of other methods.

## 3.5 Comparison with regression methods

The prediction power of CDSML was further compared to the power of four state-of-the-art regression models: SRMF [6], CaDRReS [7], CDCN [8], and ADRML [9]. The implementation of these methods were available. We applied $C_{max}$ threshold on the predicted IC50 values by these methods and convert them to the classification models. The performance of these models are provided in Table 9. On GDSC datset, CaDRReS achieved reasonable results. Moreover, SRMF and CDCN showed weak performance. Meanwhile, CDSML performs better than these methods with regard to all criteria. The interesting part of this evaluation is the comparison

**Table 7. The evaluated hyper-parameters and the best values of hyper-parameters for off-the-shelf classification methods.**

| Method | Evaluated hyper-parameters | Best hyper-parameters |
|--------|---------------------------|----------------------|
| GNB | Variance smoothing:$\{10^{-12}, 10^{-9}, 10^{-6}, 10^{-3}, 10^{-1}\}$ | Variance smoothing: 0.1 |
| LR | Regularization scale:$\{ 0.001, 0.1, 1, 0.01, 10, 100\}$ | Regularization scale: 1 |
|  | Stop tolerance:$\{10^{-6}, 10^{-4}, 10^{-2}\}$ | Stop tolerance: $10^{-6}$ |
| RF | Criterion: {gini, entropy} | Criterion: entropy |
|  | Number of trees:{10,50,100,500,1000} | Number of trees: 100 |
| SVM | Kernel: {linear, poly, RBF, sigmoid, precomputed} | Kernel: linear |
|  | Regularization parameter: {0.01,0.1,1,10,100} | Regularization parameter: 0.1 |
| MLP | Hidden layer sizes: {(50,50,50), (50,100,50), (100,)} | Hidden layer sizes: (50,50,50) |
|  | Activation function: {tanh, ReLU} | Activation function: ReLU |
|  | Solver: {SGD, Adam} | Solver: Adam |
|  | Learning rate: {Constant, Adaptive} | Learning rate: Adaptive |
|  | Regularization term: {0.0001, 0.05} | Regularization term: 0.05 |
| ADA | Number of estimators: {10,50,100,500,1000} | Number of estimators: 50 |
|  | Learning rate: {1,1.25,1.5,1.75,2} | Learning rate: 1.25 |
| KNN | K: {3,5,7,9,11,13,15,17,19,21,23,25} | K: 19 |
| CDSML | $\alpha, \beta \in \{0.125, 0.25, 0.5, 1, 1.5, 2, 2.5, \cdots, 8\}$ | $\alpha = 3.5, \beta = 4.5$ |
|  | $K = k' * \min(m, n)$, where $k' \in \{0.1, 0.2, \cdots, 0.9\}$ | $k' = 0.7$ |

**Table 8. Comparison of CDSML's performance with off-the-shelf methods' performance on GDSC and CCLE.** The assessments were done by averaging 30 repetitions of stratified five-fold cross-validation. The highest value of each criterion is shown in bold.

| Dataset | Method | AUC | AUPR | Accuracy | F1-score | Precision | Recall |
|---------|--------|------|------|----------|----------|-----------|--------|
| GDSC | CDSML | **0.9157** | 0.9398 | **0.8388** | **0.8715** | 0.8422 | **0.9031** |
| GDSC | GNB | 0.8662 | 0.8974 | 0.7792 | 0.8033 | 0.8695 | 0.7465 |
| GDSC | LR | 0.9037 | 0.934 | 0.8243 | 0.8524 | 0.865 | 0.8401 |
| GDSC | RF | 0.9163 | **0.9435** | 0.8376 | 0.8653 | **0.8666** | 0.8641 |
| GDSC | SVM | 0.9038 | 0.9343 | 0.8243 | 0.8522 | 0.8663 | 0.8387 |
| GDSC | MLP | 0.9044 | 0.9351 | 0.8226 | 0.8533 | 0.8529 | 0.8562 |
| GDSC | Ada | 0.9039 | 0.9346 | 0.8268 | 0.8553 | 0.8629 | 0.848 |
| GDSC | KNN | 0.9091 | 0.9371 | 0.8341 | 0.8623 | 0.8646 | 0.86 |
| CCLE | CDSML | **0.9514** | **0.977** | **0.8989** | **0.9201** | **0.9485** | **0.8934** |
| CCLE | GNB | 0.9111 | 0.9523 | 0.842 | 0.8779 | 0.8914 | 0.8675 |
| CCLE | LR | 0.9435 | 0.9651 | 0.8811 | 0.9189 | 0.948 | 0.8916 |
| CCLE | RF | 0.9494 | 0.9632 | 0.8884 | 0.9193 | 0.9463 | 0.8939 |
| CCLE | SVM | 0.9444 | 0.9694 | 0.8867 | 0.9192 | 0.9478 | 0.8924 |
| CCLE | MLP | 0.9376 | 0.9671 | 0.8744 | 0.9146 | 0.9624 | 0.872 |
| CCLE | Ada | 0.9475 | 0.9609 | 0.8878 | 0.9118 | 0.9332 | 0.8924 |
| CCLE | KNN | 0.936 | 0.9653 | 0.8788 | 0.9116 | 0.907 | 0.8843 |

between CDSML and ADRML. According to Table 9 CDSML significantly outperforms ADRML. Since CDSML uses binary response matrix, it can be inferred that using binary response matrix as the initial input of manifold learning leads to more reliable classification of cell line-drug pairs into sensitive/resistant categories.

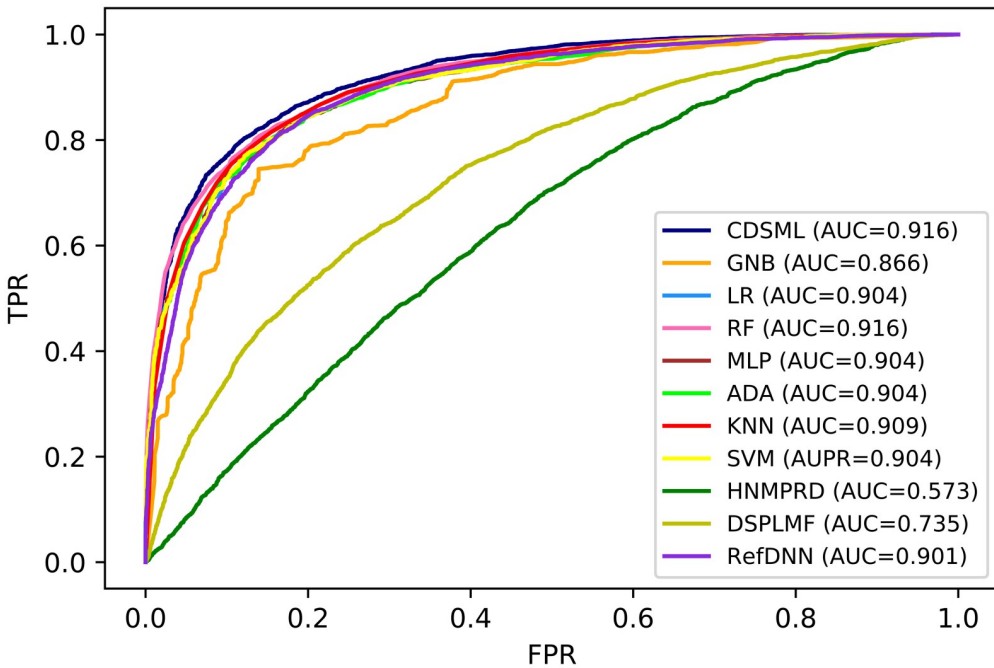

**Fig 5. The ROC curve of all classification methods on GDSC dataset.**

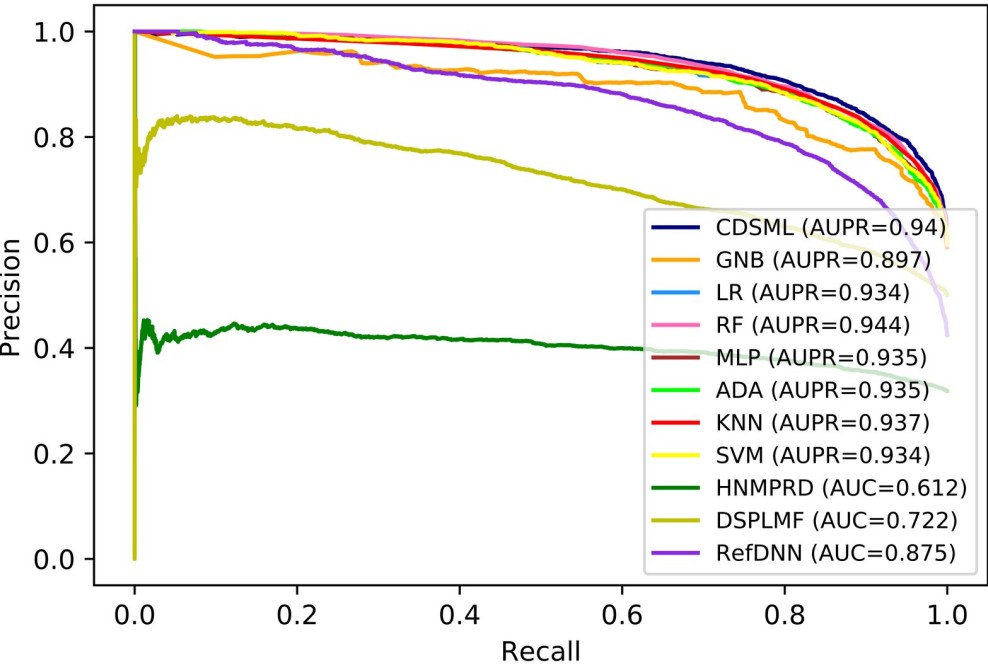

**Fig 6. The Precision-Recall curve of all classification methods on GDSC dataset.**

## 3.6 Tissue-specific conditions

Until now, we evaluated the methods on the whole dataset, which contains 19 tissue types. Nevertheless, most oncological treatments are designed based on tissue types [44], suggesting that considering tissue type may have a large impact on drug response predictions [3, 45]. Hence, we conducted tissue-specific assessments on three major tissue types.

The number of cell lines in each tissue type is shown in Fig 7. The most major tissue types are lung NSCLC, orogenital system, and leukemia with 67, 60, and 44 cell lines, respectively. We considered these tissue types and developed the tissue-specific models. The used cell lines in the train and test set for tissue-specific models belong to the same tissue type.

Figs 8–10 illustrates the predictive performance of classification methods on three major tissues. The ranking of methods performance is similar to the methods learned on the whole

**Table 9. Comparison of CDSML's performance with state-of-the-art regression methods' performance on GDSC and CCLE.** The assessments were done by averaging 30 repetitions of stratified five-fold cross-validation. The highest value of each criterion is shown in bold.

| Datset | Method | AUC | AUPR | Accuracy | F1-score | Precision | Recall |
|--------|--------|-----|------|----------|----------|-----------|--------|
| GDSC | CDSML | **0.9157** | **0.9398** | **0.8388** | **0.8715** | **0.8422** | 0.9031 |
| GDSC | SRMF | 0.4452 | 0.5493 | 0.712 | 0.7922 | 0.6908 | 0.9285 |
| GDSC | CaDRReS | 0.500 | 0.5922 | 0.6962 | 0.7738 | 0.6912 | 0.8788 |
| GDSC | CDCN | 0.4276 | 0.5460 | 0.7632 | 0.8164 | 0.7538 | 0.8903 |
| GDSC | ADRML | 0.4077 | 0.5096 | 0.7501 | 0.8177 | 0.7189 | **0.9481** |
| CCLE | CDSML | **0.9514** | **0.977** | **0.8989** | **0.9201** | 0.9485 | **0.8934** |
| CCLE | SRMF | 0.4539 | 0.6266 | 0.6966 | 0.8005 | 0.6998 | 0.9351 |
| CCLE | CaDRReS | 0.4389 | 0.6732 | 0.7133 | 0.8138 | 0.7048 | 0.9625 |
| CCLE | CDCN | 0.4262 | 0.6204 | 0.8608 | 0.8824 | **0.9796** | 0.8029 |
| CCLE | ADRML | 0.4257 | 0.6229 | 0.5473 | 0.6703 | 0.6373 | 0.7071 |

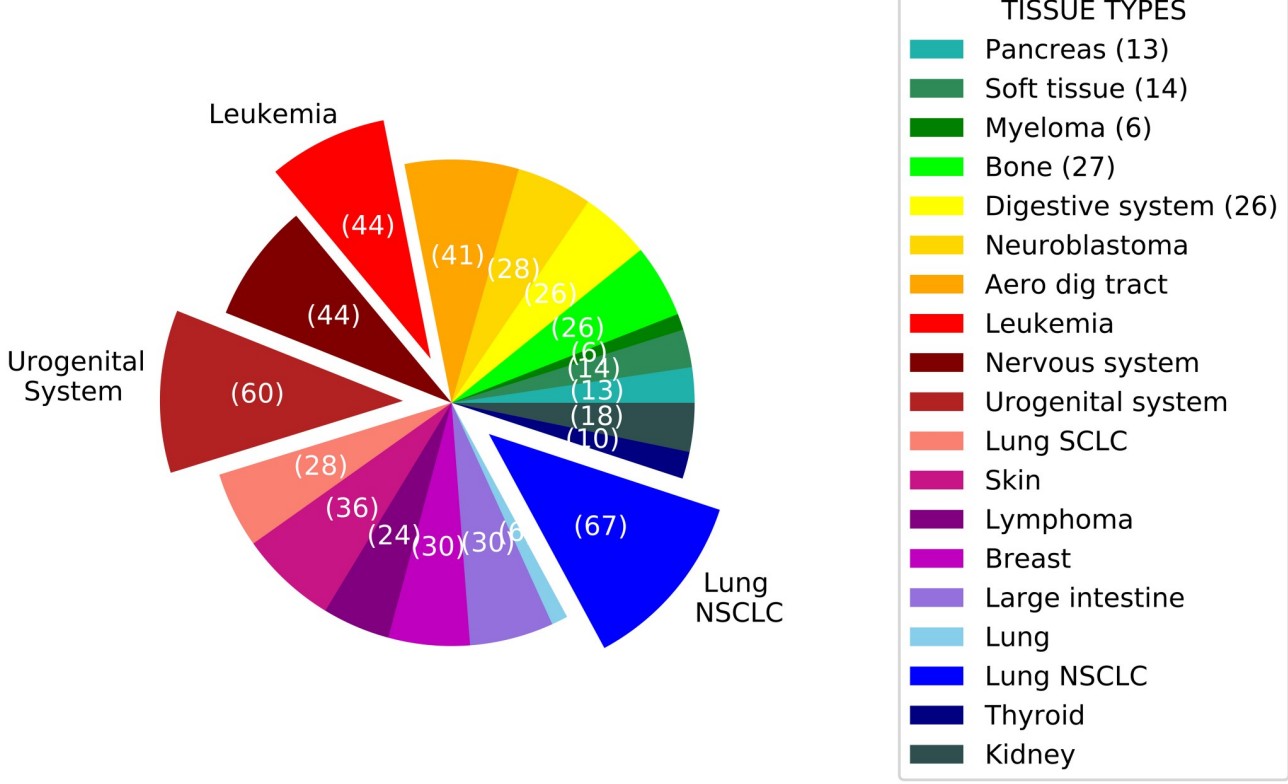

**Fig 7. The number of cell lines in each tissue type.** The number shown on slices is the number of cell lines in the related tissue type. Three major tissue types are the offset slices.

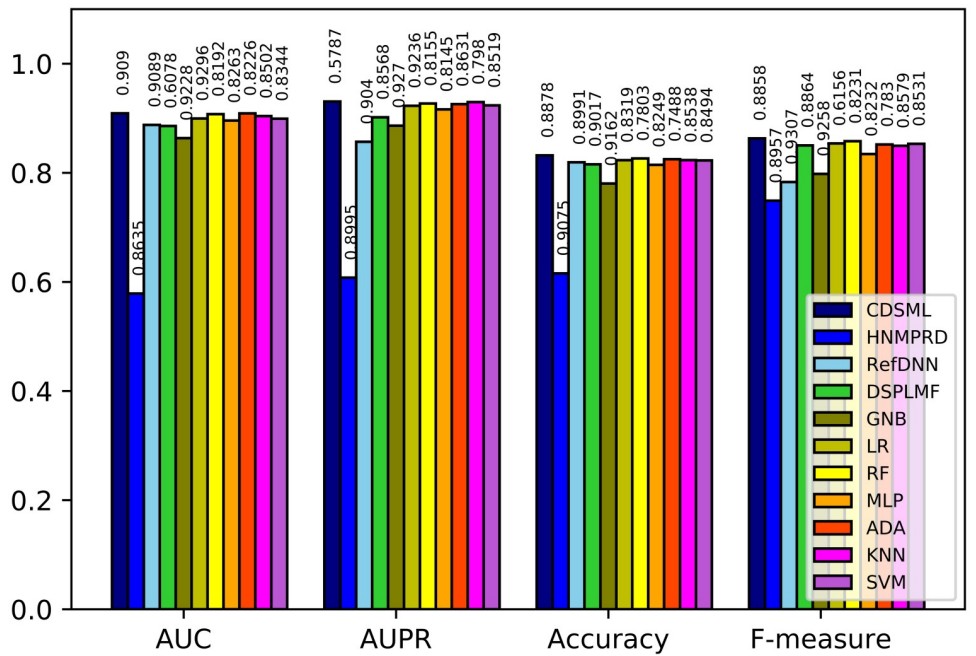

**Fig 8. The performance of methods in on NSCLC tissue.**

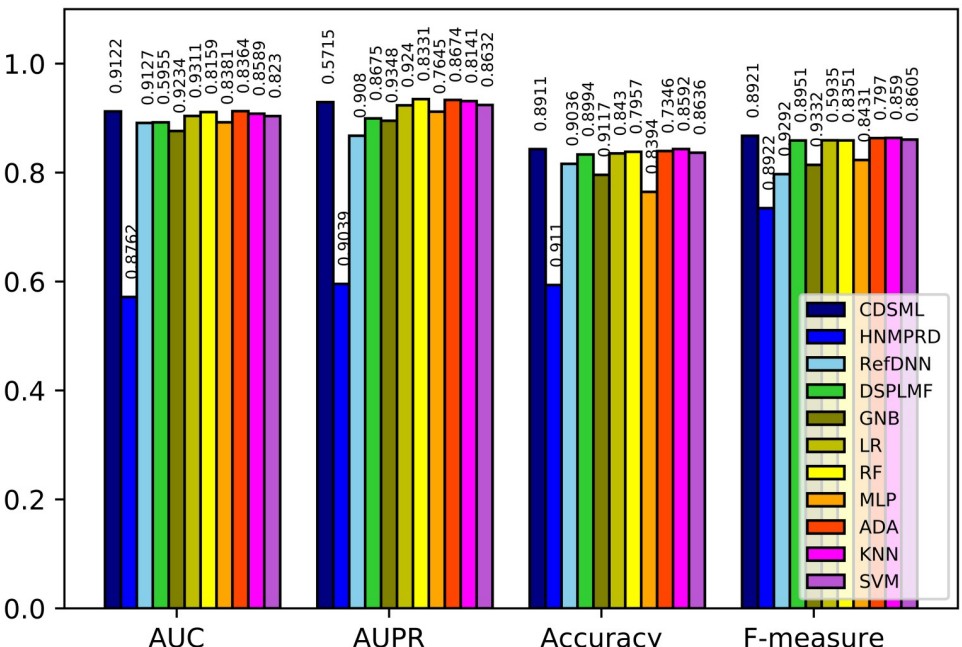

**Fig 9. The performance of methods in on orogenital system tissue.**

dataset. All methods' performance decline slightly due to the reduction in sample size, because the reduction in sample size limits the predicted power of models [46]. Nevertheless, CDSML outperforms other methods in three tissue-specific scenarios. The *AUC* of CDSML were 0.9096, 0.9186, and 0.9106 on leukemia, urogenital system, and NSCLC tissues, receptively.

### 3.7 Case studies

As it was mentioned, some cell line-drug pairs have missing IC50 values, which were imputed in the pre-processing step. In order to conduct case studies, we considered the predictions for missing pairs and investigated their predicted novel sensitive pairs. There *B* matrix had 7790 missing values which accounts for roughly 40% of all samples. The predicted sensitivity scores for these pairs were obtained and sorted. The list of top 2000 most sensitive and top 2000 most resistant cell line-drug pairs are provided in S7 and S8 Tables in S1 File, respectively.

The top 15 most sensitive pairs that had missing associations in the original dataset were considered for further analysis. Table 10 represents the list of top 15 ranked samples assigned as sensitive. Reliable literature and the latest version of GDSC database were probed to provide evidence for the novel sensitive associations. As it is shown in Table 10, all top 15 novel associations were verified as sensitive pairs in the final version of GDSC. In addition, there are numerous insights about these associations in the literature.

These associations mainly report sensitive cell lines for Ponatini, VX-7002, Temsirolimus, Lenalidomide, Vinorelbine, Epothilone B, Docetaxel, among which, the insights about sensitive associations of three drugs are described in the following.

Ponatinib is a tyrosine kinase inhibitor which hinders the activity of four FGFR [63]. A recent study have shown that inhibits the cell growth in cell lines of various tissues types such as colon cancer [47]. Researchers states that the multi kinase inhibitors such as Ponatinib have showed efficient activity in targeting pancreatic cancer cells [49]. In addition, its effectiveness

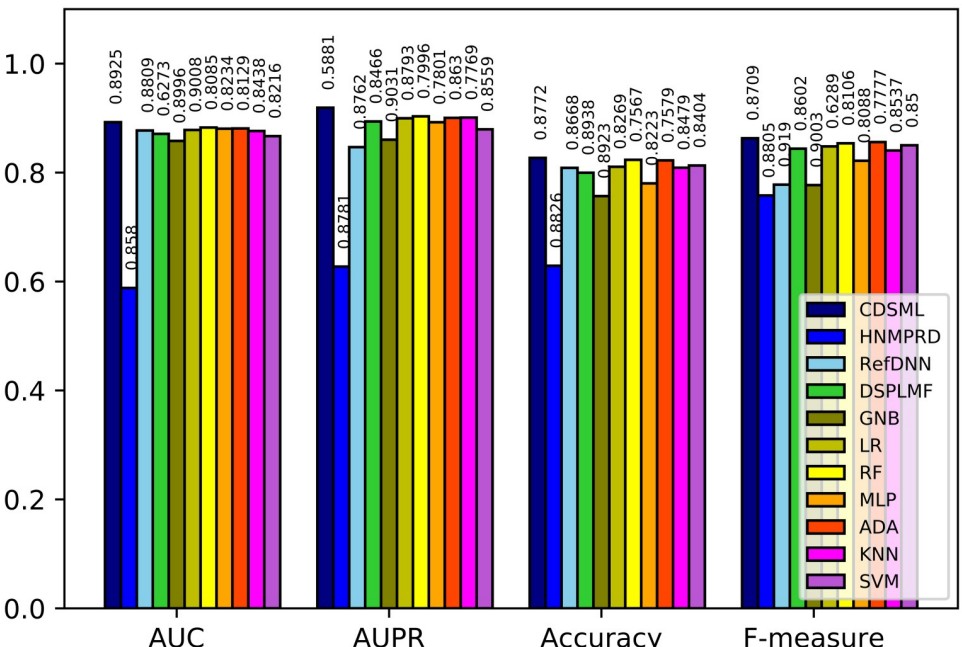

**Fig 10. The performance of methods in on leukemia tissue.**

in the treatment of cell lines with head and neck cancer have been evaluated in several studies [47, 62].

VX-702—a P38 mitogen-activated protein kinase inhibitors—have been developed for the treatment of the inflammation diseases [64]. P38$\beta$ MAPK is highly expressed in lung tissues and P38 MAPK pathways are highly activated in SCLC and breast cell lines, which leads to tumorgenesis and metastasis [48, 50]. The oral treatment with VX-702 seems to be effective in diminishing the fibrosis in SCLC and breast [50]. HuangFu *et al.* have shown that the

**Table 10. Top 15 novel sensitive pairs and pieces of evidence for these pairs.**

| Rank | Cell line name | Drug name | Cell line tissue | Sensitivity Score | Literature evidence | GDSC verification |
|------|----------------|-----------|------------------|-------------------|---------------------|-------------------|
| 1 | CW-2 | Ponatinib | large intestine | 0.9253 | [47] | verified |
| 2 | ZR-75-30 | VX-702 | breast | 0.9248 | [48] | verified |
| 3 | YAPC | Ponatinib | pancreas | 0.9062 | [49] | verified |
| 4 | NCI-H1092 | VX-702 | SCLC | 0.9017 | [50, 54] | verified |
| 5 | NCI-H1092 | Temsirolimus | SCLC | 0.8993 | [51] | verified |
| 6 | CW-2 | Vinorelbine | SCLC | 0.8929 | [52] | verified |
| 7 | NCI-H1563 | VX-70 | NSCLC2 | 0.8823 | [53] | verified |
| 8 | COR-L88 | VX-702 | SCLC | 0.8818 | [50, 54] | verified |
| 9 | SHP-77 | VX-702 | SCLC | 0.8783 | [50, 54] | verified |
| 10 | LB373-MEL-D | Lenalidomide | melanoma | 0.8758 | [55, 56, 57] | verified |
| 11 | COR-L88 | Temsirolimus | SCLC | 0.8753 | [51] | verified |
| 12 | CP66-MEL | VX-702 | melanoma | 0.875 | [58] | verified |
| 13 | CW-2 | Epothilone B | large intestine | 0.8738 | [59] | verified |
| 14 | NCI-H1092 | Docetaxel | SCLC | 0.8648 | [60, 61] | verified |
| 15 | SCC-9 | Ponatinib | head and neck | 0.8619 | [47, 62] | verified |

administration of VX-702 along with *INFβ* treatments from melanoma, stabilization of the cell response, and improvement in the treatment efficacy [58].

Lenalidomide which has tumoricidal and immunomodulatory roles, has been frequently studied for the treatment of malignant melanoma and leads to high efficiency in combination with Docarbazine [55–57].

The supportive pieces of evidence in literature and GDSC database verified that CDSML could efficiently predict drug response label associations for the cell-line drug pairs.

## 4 Conclusion

In this study, we proposed CDSML, a classification method for predicting anti-cancer drug sensitivity by applying manifold learning. It applies four steps of pre-processing, namely, imputing missing values, converting IC50 values to binary labels using max concentration thresholds, similarity calculations, standardization and normalization. We used an imputation procedure to fill the missing values. The similarities of drugs were computed based on the chemical substructure of drugs, STiTCH network, and PPI of drug targets. We considered three types of cell line similarities based on gene expression, mutation, and copy number variation of cell lines.

We extended the CDSML application, so that it can be performed on missing values without imputation. Moreover, CDSML can be performed in three similarity settings: using no similarity information, using only one of the cell line or the drug similarities, and using both cell line and drug similarities. CDSML shows high performance in all similarity settings. Even when no similarity is used for training the model, CDSML succeeds in achieving favorable results and outperform many of state-of-the-art methods. When CDSML uses only one of the cell line or the drug similarities, it makes about 6% improvement compared to the case of not using any similarities. However, making use of both cell line and drug similarities make subtle improvement. Additionally, the performance of CDSML is robust on different types of similarities. In other words, making use of various types of similarities by CDSML leads to accurate and almost similar results.

We conducted several validations to assess the rationality of imputation procedure. To this aim, we compared the imputed values with predictions of another state-of-the-art method. In another validation, we replace the suggested implementation procedure with another method. All of validations confirmed that the suggested implementation procedure fills the missing values with reasonable values and using this procedure leads to more reliable results.

For comparison of CDSML performance, we compared its results with three sets of methods: state-of-the-art classification methods, off-the-shelf classification machine learning approaches, and state-of-the-art regression methods. The methods considered in comparisons cover diverse methodologies. In order to compare the results of CDSML with regression methods, we applied max concentration threshold on the predicted IC50 for converting them to sensitive/resistance labels. The methods were evaluated by averaging common classification criteria over 30 repetitions of stratified five-fold cross-validation. The higher performance of CDSML than other methods verifies its efficient predictive power.

We further compared CDSML performance in tissues-specific conditions, because tissue-type may influence the drug response. To this aim, we considered three major tissue types including NSCLC, urogenital system and leukemia. Then, we trained the models on each tissue type. The predictive performance of methods decline subtly on tissue-specific data due to the reduction in sample size. However, CDSML achieved better results in all tissue-specific scenarios, which suggest its capability in retrieving drug sensitivity for each tissue type.

Some of drug responses were unknown in the original dataset. The predicted sensitive associations for the Unknown pairs were considered as case studies and investigated in the latest version of GDSC along with reliable literature. All top 15 novel sensitive predicted pairs were verified in the GDSC database and several pieces of evidence support the novel associations. Therefore, the performance of CDSML in predicting anti-cancer drug sensitivity is efficient.

Some of contributions of this paper are listed below:

- The idea of CDSML was inspired from ADRML, which was a regression method and uses IC50 values for training the model. We changed ADRML in such a way that it can be used in the classification form with high efficiency. The results evidently showed that performing ADRML and just applying a threshold on the predicted IC50 values does not lead to satisfactory performance in classification area. Since suggesting efficient drugs for patients in precision medicine uses the sensitivity of patients to the anti-cancer drugs, it is more essential to predict the sensitivity or resistance label instead of the response values. Therefore, the classification problem has a higher importance than regression problem in this area. In this paper, we have shown that applying max concentration threshold on the inputs and predicting sensitive/resistant labels using manifold learning leads to more reliable sensitivity prediction.

- Moreover, it is helpful to figure out the importance of each cell line or drug similarities on the performance of the classification model. In this paper, we thoroughly investigate the effect of using no similarity, using only the cell line similarity, using only the drug similarity, and using both the drug and cell line similarities on the prediction performance.

- Numerous previous studies have proposed efficient methods for predicting drug responses using regression or classification models. It is highly efficient to set up a framework for comparing all regression and classification methods in a common setting and using fair comparison. We provide a framework to convert the regression models into classification models and compare all methods in classification mode. To this aim, we applied max concentration threshold on the predicted IC50 values in order to convert the predicted values into predicted sensitivity/resistance label.

- The proposed method in this paper has the capability of handling missing values with or without imputation strategy. Moreover, the implemented code is able to perform the proposed method using no similarity, one type of similarity, or both cell line and drug similarities. The implemented code has the capability to adopt the suitable loss function and optimization procedure based on the options that the user determined for the usage of similarity information.

- Several previous papers have used an imputation approach based on nearest neighbors to impute the missing values in response matrix or the feature matrices. The reasonability of using this procedure for imputing missing values were not fully validated in the previous studies. Here, we confirmed the reliability of using this procedure for imputing missing values using four different scenarios.

- Since the algebraic and spectral characteristics of matrices used in manifold learning influence the convergence of the model, we proposed to use the combination of standardization and normalization in this paper in order to handle the negative similarity values and transforming the similarities to more informative matrices with desirable characteristics. Using the combination of standardization and symmetric Laplacian normalization is novel.

- Furthermore, we computed two novel similarities for drugs based on the maximum matching in target PPI network obtained from STRING database and the computed Jaccard index

in STiTCH network. These two types of similarities for drugs are fully described in the response of subsequent comments and in the revised manuscript. It should be noted that the proposed idea for computing these drug similarities leads to the calculation of highly informative and comprehensive similarities for drugs using the combination of various types of information. Since several drugs in GDSC are not FDA-approved the information about these drugs are not rich. Therefore, introducing new similarities for these anti-cancer drugs is productive. The newly introduced drug similarities may help future studies in this field and other fields related to drug discovery.

## Supporting information

**S1 File.**
(XLSX)

**S2 File.**
(PDF)

## Acknowledgments

The authors would like to thank Narjes Rohaniher assistance in implementations and assessments.

## Author Contributions

**Conceptualization:** Fatemeh Ahmadi Moughari, Changiz Eslahchi.

**Formal analysis:** Fatemeh Ahmadi Moughari.

**Investigation:** Fatemeh Ahmadi Moughari.

**Methodology:** Fatemeh Ahmadi Moughari.

**Supervision:** Changiz Eslahchi.

**Validation:** Fatemeh Ahmadi Moughari.

**Visualization:** Fatemeh Ahmadi Moughari.

**Writing – original draft:** Fatemeh Ahmadi Moughari.

**Writing – review & editing:** Fatemeh Ahmadi Moughari, Changiz Eslahchi.

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
