## [Decision Letter · Decision Letter 0]

9 Feb 2021

PONE-D-20-32564

A Computational Method for Drug Sensitivity Prediction of Cancer Cell Lines Based on Various Molecular Information

PLOS ONE

Dear Dr. Eslahchi,

Thank you for submitting your manuscript to PLOS ONE. After careful consideration, we feel that it has merit but does not fully meet PLOS ONE’s publication criteria as it currently stands. Therefore, we invite you to submit a revised version of the manuscript that addresses the points raised during the review process.

We look forward to receiving your revised manuscript.

Kind regards,

Fatemeh Vafaee, Ph.D.

Academic Editor

PLOS ONE

Journal Requirements:

2. Please upload a new copy of Figure 4 as the detail is not clear. Please follow the link for more information: https://blogs.plos.org/plos/2019/06/looking-good-tips-for-creating-your-plos-figures-graphics/" https://blogs.plos.org/plos/2019/06/looking-good-tips-for-creating-your-plos-figures-graphics/

Reviewers' comments:

Reviewer's Responses to Questions

**Comments to the Author**

1. Is the manuscript technically sound, and do the data support the conclusions?

Reviewer #1: Yes

Reviewer #2: Yes

2. Has the statistical analysis been performed appropriately and rigorously? 

Reviewer #1: Yes

Reviewer #2: Yes

3. Have the authors made all data underlying the findings in their manuscript fully available?

Reviewer #1: Yes

Reviewer #2: Yes

4. Is the manuscript presented in an intelligible fashion and written in standard English?

Reviewer #1: Yes

Reviewer #2: No

5. Review Comments to the Author

Reviewer #1: The authors introduced the CDSML method to predict drug sensitivity of cancer cell lines. They compared it with two types of methods – classification and regression methods on the GDSC dataset. It shows the CDSML performs better than other methods when using the six metrics. This manuscript is well organized and easy to follow. It provides us an effective computational method for drug sensitivity prediction on cancer cell lines. I have the following comments.

Major:

1. In the objective function, it seems that you could solve this problem without dealing with missing values. Because in the first term in Eq.(8), one could ignore the drug-cell line pair for which B(i,j) value is missing.

2. Following the above question, could the author compare the predicted values for missing pairs and the imputing values by Eq.(1) to show the reasonability of the imputing method in the paper.

3. The idea of the method in this paper ”Predicting Anticancer Drug Responses Using a Dual-Layer Integrated Cell Line-Drug Network Model” is very similar to the imputing way (Eq.(1)) for missing values, could you compare CDSML with it?

4. Regarding to the objective function, how to ensure the entries in the estimated matrix B range from 0 to 1? There is no constraint about the scale and sign for values of X(i) and Y(j).

5. In Table 2, CDSML performs similarly when using gene expression, mutation and CNV to calculate cell line similarity. How about estimating B(i,j) without the matrix SC in the objective function? It will test the power of cell line similarity matrix SC and drug similarity matrix SD when prediction.

6. When performing the 5-fold CV procedure, did you make sure that the percentage of positive and negative cases for training dataset is the same with the testing dataset? I think this is important for classification problem. Under your threshold C_{max}, please provide the percentage of sensitive drug-cell line pairs.

7. CCLE is another commonly used dataset for drug sensitivity prediction study. It is better to apply the methods on this dataset to show its efficiency.

Minor:

1. In the Abstract, “achieving AUC of 0.92%, xxxxx 0.94%, xxxx”. The percent sign % should be removed.

2. Please check the compared method names, and make sure they are the same in the manuscript, for example, SRMF and SRMf.

3. Could you provide the values for the parameters \\alpha, \\beta and the latent number k in Eq.(8) in the Table 4? And in Table 4, which metric do you use to select the best hyper-parameters for these methods since there are 6 metrics in Table 5?

4. The figures in this paper have very low resolution, please improve that.

5. The section number is strange, 0.1, 0.2 ….

6. I downloaded the data from the github website, all the entries in the table “SC_CopyNumberVariation.csv” are zero. Did you upload a wrong table?

Reviewer #2: 1. Avoid using abbreviation in abstract

2. Performance gain in the new method (CDSML) compared to the old one (ADRML) should be made clear

3. Author should clarify what is the novelty in this work compared to the previous approach (i.e. ADRML).

4. Writing should be improved in several places, e.g. several short sentences should be paraphrased in a better way.

5. Data preprocessing steps should be clearly mentioned in the method section

6. Including two types of cell-line similarity profiles, authors have used only one type of drug similarity profiles based on chemical substructure. I would suggest to include other types of multi-modal drug similarities, e.g. target protein sequence similarity, target pathway similarity and functional similarity would affect the overall perfomance. See Azad et. al, 2020 [Brief. in Bioinf.] for reference.

7. Author need to merge 0.5.1.c and 0.5.2

8. Author should provide experimental justification for the threshold of 0.6 at line 177.

9. In the case studies, literature evidence should be sought for the predicted drug-cell line associations, not just drug-tissue association in general.

10. More detail discussion should be made about the methodologies, results found and various assumptions.

6. PLOS authors have the option to publish the peer review history of their article (what does this mean?). If published, this will include your full peer review and any attached files.

Reviewer #1: No

Reviewer #2: No

---

## [Author Response · Author response to Decision Letter 0]

30 Mar 2021

The authors would like to thank the reviewers for their deep and accurate review. Following the reviewers’ comments, we have included all the comments point by point, made the required changes in the manuscript, and highlighted them. We hope the revised manuscript will better suit the "PLOS One Journal". The attached file is our detailed response to the reviewers’ remarks.

---

## [Decision Letter · Decision Letter 1]

12 Apr 2021

A Computational Method for Drug Sensitivity Prediction of Cancer Cell Lines Based on Various Molecular Information

PONE-D-20-32564R1

Dear Dr. Eslahchi,

We’re pleased to inform you that your manuscript has been judged scientifically suitable for publication and will be formally accepted for publication once it meets all outstanding technical requirements.

Kind regards,

Fatemeh Vafaee, Ph.D.

Academic Editor

PLOS ONE

Additional Editor Comments (optional):

Reviewers' comments:

Reviewer's Responses to Questions

**Comments to the Author**

1. If the authors have adequately addressed your comments raised in a previous round of review and you feel that this manuscript is now acceptable for publication, you may indicate that here to bypass the “Comments to the Author” section, enter your conflict of interest statement in the “Confidential to Editor” section, and submit your "Accept" recommendation.

Reviewer #1: All comments have been addressed

Reviewer #2: All comments have been addressed

2. Is the manuscript technically sound, and do the data support the conclusions?

Reviewer #1: Yes

Reviewer #2: Yes

3. Has the statistical analysis been performed appropriately and rigorously? 

Reviewer #1: Yes

Reviewer #2: Yes

4. Have the authors made all data underlying the findings in their manuscript fully available?

Reviewer #1: Yes

Reviewer #2: Yes

5. Is the manuscript presented in an intelligible fashion and written in standard English?

Reviewer #1: Yes

Reviewer #2: Yes

6. Review Comments to the Author

Reviewer #1: (No Response)

Reviewer #2: All comments are rigorously addressed.

7. PLOS authors have the option to publish the peer review history of their article (what does this mean?). If published, this will include your full peer review and any attached files.

Reviewer #1: No

Reviewer #2: **Yes: **A K M Azad

---

## [Editor Report · Acceptance letter]

19 Apr 2021

PONE-D-20-32564R1 

A Computational Method for Drug Sensitivity Prediction of Cancer Cell Lines Based on Various Molecular Information  

Dear Dr. Eslahchi:

I'm pleased to inform you that your manuscript has been deemed suitable for publication in PLOS ONE. Congratulations! Your manuscript is now with our production department. 

Kind regards, 

on behalf of

Dr. Fatemeh Vafaee 

Academic Editor

PLOS ONE